# A Rolling Bearing Fault Diagnosis Method Based on EEMD-WSST Signal Reconstruction and Multi-Scale Entropy

**DOI:** 10.3390/e22030290

**Published:** 2020-03-02

**Authors:** Jianghua Ge, Tianyu Niu, Di Xu, Guibin Yin, Yaping Wang

**Affiliations:** Key Laboratory of Advanced Manufacturing and Intelligent Technology, Ministry of Education, Harbin University of Science and Technology, Harbin 150080, China; gejianghua0619@gmail.com (J.G.); rookiety1015@gmail.com (T.N.); dixu115@hrbust.edu.cn (D.X.); aarongrantyin@gmail.com (G.Y.)

**Keywords:** rolling bearing, fault diagnosis, EEMD, wavelet semi-soft threshold, multi-scale entropy

## Abstract

Feature extraction is one of the challenging problems in fault diagnosis, and it has a direct bearing on the accuracy of fault diagnosis. Therefore, in this paper, a new method based on ensemble empirical mode decomposition (EEMD), wavelet semi-soft threshold (WSST) signal reconstruction, and multi-scale entropy (MSE) is proposed. First, the EEMD method is applied to decompose the vibration signal into intrinsic mode functions (IMFs), and then, the high-frequency IMFs, which contain more noise information, are screened by the Pearson correlation coefficient. Then, the WSST method is applied for denoising the high-frequency part of the signal to reconstruct the signal. Secondly, the MSE method is applied for calculating the MSE values of the reconstructed signal, to construct an eigenvector with the complexity measure. Finally, the eigenvector is input to a support vector machine (SVM) to find the fault diagnosis results. The experimental results prove that the proposed method, with a better classification performance, can better solve the problem of the effective signal and noise mixed in high-frequency signals. Based on the proposed method, the fault types can be accurately identified with an average classification accuracy of 100%.

## 1. Introduction

Rolling bearing plays an important role in the rotating machinery as it has a pointedly effect on its service life. Among the causes of abnormalities in rotating machinery, 30% of abnormalities are caused by bearing failure. Hence, the way for quick and effective fault diagnosis has been one of the emphases of research among scholars worldwide. When a rolling bearing failure occurs, its vibration signal changes, and its fault information is contained in these changed vibration signals. Under increasingly complex working conditions for rotating machinery, it is difficult to diagnose faults in rolling bearing with only individual subjective experiences. Hence, signal analysis is a necessarily process for accurate fault diagnosis. Meanwhile, the vibration signal of bearings often contains abundant noise signals due to the complex working conditions, which adds a great deal of difficulties to the failure form and performance prediction of rolling bearings, affecting the accuracy of judgment. Hence, noise reduction is quite vital and meaningful before the diagnosis [1].

In addition to the frequently-used time domain [2] and frequency domain analysis [3], current signal denoising methods have also developed some very advanced time-frequency analysis methods, such as empirical mode decomposition (EMD) [4], blind source separation (BSS) [5], energy entropy [6], variational mode decomposition (VMD) [7], wavelet transformation (WT) [8], approximate entropy (ApEn) [9], etc. Nonlinear and nonstationary signals can be decomposed into multiple intrinsic mode functions (IMFs) by EMD [10]. Aiming at the shortcomings of the EMD, such as modal mixture and end effect, which affect the accuracy of signal decomposition [11], Wu et al. [12] improved the EMD method with auxiliary noise and proposed the ensemble empirical mode decomposition (EEMD) method. 

The EEMD method results in signal decomposition with anti-noise characteristics, reduces reconstruction errors, and improves the quality of IMFs by adding Gaussian white noise to the original signal. Therefore, the EEMD method is widely used in signal processing and fault diagnosis [13]. However, when the EEMD method is applied independently for decomposing signals and reducing noise, the information in the high-frequency components is also lost as some IMFs are discarded [14]. The WT method has a good performance on the suppression of random noise by having the properties of multi-scale, low entropy, and decorrelation [15]. Jumah et al. [16] proposed a method using wavelet transform and various thresholding techniques, which has a good effect on removing one-dimensional Gaussian white noise. However, there is no uniform standard for the operation of the wavelet threshold method, which has a greater impact on the final result. The wavelet hard threshold denoising method generates discontinuous points and loses some vital information [17]. The wavelet soft threshold denoising method causes distortion phenomena, such as edge blur effect [18]. 

Although the wavelet threshold denoising method effectively removes noise from high-frequency signals, the effect is not ideal, and the useful signals still mix with noise signals. Nevertheless, the wavelet semi-soft threshold with the advantages of the hard and soft threshold can not only preserve the integrity of the signal but also ensure the accuracy of noise reduction [19]. However, the bearing vibration signals are usually complexity and non-stationarity due to the genesis of failures. Therefore, the single time-frequency analysis method is restricted in extracting the features. The extracted features are not ideal for distinguishing bearing fault types.

Aiming at the problem that the bearing features masked in the nonlinear signals cannot be effectively extracted, to ensure accurate fault diagnosis, we need to research from the perspective of signal complexity. So far, many scholars have proposed many methods to characterize the complexity of signals, and entropy theory is one of them [20]. Entropy theory has been applied for mechanical fault diagnosis by resorting to effectively extracting the fault features masked in the nonlinear vibration signal [21]. 

Costa et al. [22] proposed the concept of the multi-scale entropy (MSE) method. The MSE method can measure the complexity of signals from the perspective of the similarity of the data and the self-correlation of the time series. Therefore, the MSE method is employed to extract features and analyze the irregular degree of the signal on different time scales. Zheng et al. [23] proposed a method, which uses MSE to measure the complexity of rotor fault signals for fault feature extraction and to accurately extract the difference information from different fault signals. 

Tiwari et al. [24] proposed a method combining multi-scale permutation entropy (MPE) with the adaptive neural fuzzy classifier (ANFC), which can diagnose bearing fault information and predict early bearing failure. Rodriguez et al. [25] proposed a method with multi-scale wavelet entropy (MWE) combined with the kernel limit learning machine (KELM); however, they only considered experimental data and did not verify more complex practical data pretreatment. 

With the increasing complexity of today’s mechanical equipment and working environments, where rolling bearings work under hostile conditions, the vibration signals are often more complex and irregular. Therefore, it is difficult to directly use the entropy methods to extract fault features that can effectively distinguish fault types. For this problem, some scholars preprocessed the original vibration signals before using entropy theory for feature extraction. Hsieh et al. [26] utilized the EMD and the MSE for high-speed spindle fault diagnosis. Their conclusion illustrates not only that MSE can accurately distinguish the fault types of high-speed spindles but also that the noise reduction performance of EMD still needs to be improved. 

Aouabdi et al. [27] used MSE and principal component analysis (PCA) to analyze current signals to monitor and diagnose the degradation of the gear. Their conclusion illustrates that this method can detect gear tooth erosion better. However, there is a deviation in the evaluation of the complexity of the sampled signal, and PCA cannot well retain the real information of the original signal. Ge et al. [28] employed multi-scale displacement entropy (MDE) and robust PCA for rolling bearing fault diagnosis, which can effectively locate and diagnose bearing faults. However, the feature components of the acquired signal are more complex than the analog signals, and the noise reduction performance requires improvement.

In summary, to obtain higher accuracy in rolling bearing fault diagnosis under complex working conditions, a method based on signal reconstruction and the MSE is presented in this paper. First, the EEMD is applied to decompose the vibration signal. Secondly, the WSST is used to denoise the modal components in the high-frequency signal filtered by correlation analysis, to reconstruct the signal. By using the EEMD-WSST method, the shortcoming of EEMD can be avoided and all the advantages of these two methods can be preserved. Different from the conventional signal processing method, the noise in the reconstructed signal is expertly filtered on the basis of ensuring the integrity of the effective part in the high-frequency signal, and this improves the efficiency of the noise reduction method. Finally, based on the effectively denoising, the MSE method is used for calculating the MSE values of the reconstructed signal. To distinguish the fault types of bearings, the calculated sample entropy is input as a rolling bearing fault feature into an SVM model, which is more suitable for training small sample data [27]. Compared with the conventional entropy methods for rolling bearing fault diagnosis, the proposed method is more stable and suitable for practical engineering applications.

The remainder of this paper is organized as follows: Section 2 introduces the signal reconstruction method based on the EEMD and WSST. Section 3 provides the feature extraction steps of MSE. Section 4 illustrates the detailed steps of the proposed method. Section 5 shows the contrast experiments of the proposed method. The conclusion is reached in Section 6.

## 2. Signal Reconstruction

### 2.1. The Basic Principles of the EEMD

The EMD decomposition is to process the signal smoothly to obtain multiple data sequences under different characteristic scales. These data sequences are called IMFs. Any signal can be regarded as a combination of several IMFs, and the IMFs obey the following two conditions: (1) the number of local extrema and the number of the zero-crossing points are equal or of only one difference in the whole time; (2) at any time, the two envelopes determined by the local maximum point and local minimum point have an average value of 0. 

The EEMD method is essentially an improved EMD method. It adds white noise into the signal mainly according to the characteristic that the mean value of white noise is zero and still decomposes the signals with EMD and averages the decomposition result. The more average processing times, the less impact of noise on the decomposition result. The EEMD is to perform multiple EMD decompositions on the signal added Gaussian white noise. The EEMD is to resort the statistical properties of the even distribution of the frequency of the Gaussian white noise to ensure the noise-added signal has continuity on different frequency scales, thereby reducing the mode mixing degree of the IMFs. 

The EEMD method equalizes noise based on the distribution characteristics of white noise spectral equalization, to make the frequency distribution tend to be uniform. Mode mixing is caused by the distribution points of the amplitude of different signals with white noise added. By using the characteristics of the EEMD method, the high-frequency modulation information is adaptively separated. This method not only weakens the mode mixing effect of the EMD method but also reduces the error caused by the central frequency band and filter band selection error in resonance demodulation. Let the signal be *x*(t). The specific decomposition steps are as follows:

Step 1. Set the average processing times of x(t) to M. Initial i=1,2,…M.

Step 2. Add random white noise $n_{i}(t)$ with a certain amplitude into x(t) to form a new series of signals.
(1)$$x_{i}(t)=x(t)+n_{i}(t)\;\;i=1,2,\ldots M$$

Step 3. Decompose the new signal sequence x(t) using the EMD method.
(2)$$x_{i}(t)=\sum_{j=1}^{J}c_{ij}(t)+r_{ij}(t)$$
where $c_{ij}(t)$ denotes the *j*-th IMF in the *i*-th decomposition, and $r_{ij}(t)$ denotes the residue of the *i*-th decomposition.

Step 4. Sum and average the corresponding IMFs obtained by the *M*-th decomposition to offset the noise and get the final IMFs.
(3)$$c_{j}(t)=\frac{1}{M}\sum_{i=1}^{M}c_{ij}(t)$$
where $c_{j}(t)$ denotes the *j*-th IMF obtained by the EEMD method, j=1,2,⋯,J.

Step 5. The final result of the x(t) decomposed by the EEMD method is as follows:(4)$$x(t)=\sum_{j=1}^{J}c_{j}(t)+r(t)$$
where r(t) denotes the final residue of signal decomposed by the EEMD method.

### 2.2. Signal Reconstruction Based on EEMD and WSST

The EMD and EEMD methods are essentially signal decomposition methods and do not have the function of removing or reducing noise compositions. These two methods decompose the original signal into multiple basic modal components, directly remove the basic modal components with higher noise components, and reconstruct the remaining basic modal components to realize the denoising process. However, these two methods will cause the loss of value signal information in high-frequency signals. The EEMD can better remove the background noise; however, it will cause the loss of the effective information in the high-frequency signal. Compared with EEMD, although wavelet denoising can better retain the effective information in the signal, its threshold influences the original signal denoising. Therefore, based on the wavelet denoising, the threshold function of wavelet denoising is improved, and a denoising method combining EEMD and improved wavelet denoising is applied to reduce the noise in the bearing signals.

Wavelet transform is suitable for nonlinear signal processing. Wavelet threshold denoising can perform noise reduction processing on the signal according to the different amplitude–frequency characteristics of the signal and has a strong ability to inhibit white noise. Therefore, wavelet threshold denoising is introduced into the noise reduction processing of the high-frequency signal to reconstruct the original signal. The most widely used wavelet transform in the signal denoising field is the wavelet threshold denoising. This is used, essentially, to filter the signal. The wavelet threshold method is to transform the signal to the wavelet domain; then obtain the wavelet coefficients and filter out noise; and finally, reconstruct the signal. The wavelet threshold function mainly includes the soft threshold as shown in Equation (5) and the hard threshold as shown in Equation (6).
(5)η(w)=(w−sgn(w)T)I(|w|>T)
(6)η(w)=wI(|w|>T)

As shown in Equation (6), the treatment method of the hard threshold is to keep the wavelet coefficients above the threshold unchanged and change the wavelet coefficients below the threshold to 0. However, this “guillotine” method will cause changes in the wavelet domain and lead to sudden local changes in the noise reduction results. The treatment method of the commonly used soft threshold is to change all the wavelet coefficients less than 3σ to 0 and, in a unified way, subtract 3σ from the wavelet coefficients greater than 3σ to make the wavelet coefficients smooth in the wavelet domain. Although this will cause some loss of useful information in the high-frequency signal, it eliminates the local concussion caused by the hard threshold function. 

The two traditional threshold acquisition methods each have advantages but also have damage to the signal. The hard threshold method provides better protection for the edge information of the original signal, while the soft threshold method smooths the signal edge after noise reduction. The disadvantage is that the soft threshold method causes a certain degree of distortion. It is obvious that the selection of wavelet threshold plays a decisive role in the whole wavelet threshold denoising process and determines the final effect of the signal denoising method. Therefore, a method is proposed to improve the above defects; this expression is as shown in Equation (7), where 0 <*T*_1_ < *T*_2_. 

The wavelet semi-soft threshold function combines the advantages of the two methods while cleverly avoiding their defects. The images of the three threshold functions are shown in Figure 1. All the abscissas represent the wavelet coefficients of the original signal, and the ordinates represent the wavelet coefficients subjected to thresholding. They are all dimensionless parameters.
(7)$$\eta (w)=\mathrm{sgn}(w)\frac{T_{2}(|w|-T_{1})}{T_{2}-T_{1}}I(T_{1}<|w|<T_{2})+wI(|w|>T_{2})$$

In the vibration signal, noise mostly exists in the high-frequency band. How to screen the IMF with high noise content from multiple IMFs is very important for the subsequent denoising process. Although EEMD can decompose signals by modal and effectively suppress mode mixing compared to EMD, there is no explicit method to determine the demarcation point of EEMD signal-to-noise components, and it is necessary to further distinguish between signal components and noise components. The function characteristics of the autocorrelation function of each modal component roughly determine the signal-to-noise demarcation point of the input signal. Then, the improved method is used to denoise the noise components after separation. Finally, the signal after denoising is recombined with the remaining components to obtain the final denoising result. 

This method retains the advantages of EEMD in effectively eliminating the background noise and overcoming the shortcomings of wavelet threshold denoising, which cannot completely eliminate the background noise. It also retains the advantages that wavelet threshold denoising can well retain the useful signals in the original signal and overcomes the shortcomings of EEMD forced denoising and loss of useful signals. The combination of the two achieves a better denoising effect by determining the noise component through the autocorrelation function. In this section, the Pearson correlation coefficient is used to analyze the correlation between the IMF component and the original signal to screen the IMF component.

The Pearson correlation coefficient is one of the indexes that can describe the degree of correlation between two random variables; therefore, it can be used as a standard that can accurately measure the correlation between two signals. Assume that for two random signals, X and Y, the Pearson correlation coefficient can be expressed as
(8)$$u=\frac{\sum_{i=1}^{n}(x_{i}-\bar{x})(y_{i}-\bar{y})}{\sqrt{\sum_{i=1}^{n}(x_{i}-\bar{x}{)}^{2}{\left(y_{i}-\bar{y}\right)}^{2}}}$$

As shown in Equation (8), if the value of the Pearson correlation coefficient u is larger, this indicates that the degree of similarity is greater between the two random signals X and Y. Conversely, if the value of the Pearson correlation coefficient u is smaller, this indicates that the similarity is smaller between the two random signals X and Y. The division of strong or weak relations of the Pearson correlation coefficient is shown in Table 1.

The linear relationship between the two signals in the high-noise components can be divided into the following three kinds: if 0<|u|<0.4, the linear relationship between the two signals is a weak linear correlation; if 0.4≤|u|<0.7, the linear relationship s is a significant correlation; if 0.7≤|u|<1, the linear relationship is a strong linear correlation, and the IMFs in this frequency band have a higher noise content. Therefore, the components in multiple IMFs that have a significant correlation and strong linear correlation with the original signal are subjected to secondary noise reduction to improve the signal-to-noise ratio (SNR) of the signal. By calculating the correlation coefficient between each IMF and the original signal, the IMFs with high noise components are picked for noise reduction to avoid the problem of effective signal loss during the mode decomposition and denoising.

To quantitatively evaluate the performance of the denoising method, the indexes for measuring the denoising effect generally include the SNR and the root mean squared error (RMSE). The SNR reflects the energy relationship between the signal and the noise, and the RMSE reflects the magnitude of the average energy of the noise. Generally, the method with a high SNR and a low RMSE is better than the method with a low SNR and high RMSE. The mathematical expressions of the SNR and the RMSE are expressed as Equations (9) and (10), respectively.
(9)$$\mathrm{SNR}=10\mathrm{lg}\left(\left(\sum_{t=1}^{N}s^{2}\left(t\right)\right)/\sum_{t=1}^{N}{\left[s\left(t\right)-x\left(t\right)\right]}^{2}\right)$$
(10)$$\mathrm{RMSE}=\frac{1}{N}\sum_{t=1}^{N}{\left[s\left(t\right)-x\left(t\right)\right]}^{2}$$
where N denotes the sampling number, s(t) denotes the noiseless original signal, and x(t) denotes the signal after denoising.

The EEMD-WSST signal reconstruction method can be briefly summarized as follows: 

Step 1. Decompose the original signal into the IMFs and residual components by EEMD.

Step 2. Obtain corresponding high-frequency noisy IMFs by Pearson correlation coefficient analysis and perform wavelet semi-soft threshold denoising on the high-frequency noisy IMFs.

Step 3. Perform signal reconstruction on the noiseless IMFs, other IMFs, and residual components.

This method reflects the advantages that the wavelet denoising method has for multi-resolution applications. EEMD adaptively decomposes the signals based on the original signal and, meanwhile, avoids the defects in practical applications, inhibits the mode mixing, and improves the denoising effect while ensuring signal integrity.

### 2.3. Analysis of Simulating Bearing Fault Signals

To verify the feasibility and effectiveness of the EEMD-WSST signal reconstruction method, a simulating rolling bearing signal is constructed. It can be described by Equation (11).
(11)$$s(t)=y_{0}e^{-2\pi f_{n}gt}\sin\left(2\pi f_{n}\sqrt{1-g^{2}t}\right)+n\left(t\right)$$
where s(t) denotes the noise-added rolling bearing simulation signal, $y_{0}$ is the displacement constant, $f_{n}$ is the natural frequency of the rolling bearing, t is the sampling time, g is the damping coefficient, and n(t) is the Gaussian white noise that tends to be real noise. Here, we set $y_{0}=3$, $f_{n}=3000\;\mathrm{Hz}$, g=0.09.

When s(t)−n(t), the simulated signal is pure and noiseless; the amplitude and spectrum diagrams of the noiseless simulated signal are presented in Figure 2. The time domain and frequency domain chart of the simulated signal after adding Gaussian white noise are given in Figure 3.

Figure 3 shows that the time-domain waveform becomes more confusing after adding white noise. In the frequency waveform, due to simulating a strong noise background, higher amplitudes appear at certain frequencies. It is significant to be able to extract the required information in a high-noise background; extracting useful information in a Gaussian-noise background is a technical problem. Therefore, we use this simulated signal to verify the superiority of the denoising method we proposed and show the robustness of this denoising method.

To verify that the proposed method has a better noise reduction effect compared with other traditional methods, the following methods are compared experimentally: EEMD forced denoising, wavelet threshold denoising, EEMD combined with wavelet hard threshold (EEMD-WHT), EEMD combined with wavelet soft threshold (EEMD-WST), EMD combined with wavelet semi-soft threshold (EMD-WSST), and EEMD-WSST.

Figure 4 presents the noiseless simulated signal decomposed by the EEMD. Figure 5 presents the noise-added simulated signal decomposed by the EEMD. As can be seen in Figure 4 and Figure 5, although the EEMD can adaptively decompose the original signal into multiple IMFs at different frequencies, there exists the problem of mode mixing.

The time domain and frequency domain chart of the simulated signal, after EEMD forced denoising, are presented in Figure 6. It can be seen from the frequency waveform that the EEMD forced denoising will cause the loss of high-frequency signals. The high-frequency part of the frequency waveform is almost zero. This denoising method will cause signal distortion to some extent.

The time domain and frequency domain chart of the simulated signal after wavelet threshold denoising are presented in Figure 7. In this simulation analysis, the basis function used for wavelet threshold noise reduction is db3, and the decomposition level is 3. From Figure 7, we can observe that the effect of wavelet threshold denoising is not ideal, as the wavelet threshold denoising requires artificial selection of the wavelet base function and the number of decomposition layers. The default threshold is not a suitable selection, as it causes the signals with frequencies higher than 2000 HZ to be almost filtered out and also filters out the important part of the signal.

The EEMD-WSST signal reconstruction method is based on correlation analysis. Pearson correlation analysis is performed on the original signal and the IMFs obtained by EEMD, and the wavelet threshold denoising is performed on IMF with the Pearson correlation coefficient greater than 0.7. Then, we reconstruct the signal. The correlation analysis results are shown in Table 2.

In the EEMD-WSST, the IMFs with a Pearson correlation coefficient greater than 0.7 are selected for wavelet semi-soft threshold denoising. Therefore, IMF1 and IMF2 are selected.

The time domain and frequency domain chart of the simulated signal after denoising by EEMD-WHT are shown in Figure 8. It can be seen that the EEMD-WHT can filter noisy signals to a certain extent, but the effect is not ideal. Figure 9 presents the time domain and frequency domain chart of the simulated signal after denoising by the EEMD-WST. Compared to the EEMD-WHT, the time-domain waveform obtained by the EEMD-WST is more concise. However, the EEMD-WST will cause the lower amplitude of the high-frequency part of the frequency waveform, which may submerge the effective information. 

In Figure 10, the time domain and frequency domain charts of the simulated signal denoised by the EMD-WSST are shown. In Figure 11, the time domain and frequency domain charts of the simulated signal denoised by EEMD-WSST are presented. We can observe from the time domain waveform in Figure 10 and Figure 11, at the 400–5000 HZ section, the denoising performance of the EEMD-WSST is better than the EMD-WSST. The former is closer to a pure and noiseless simulated signal at the 400–5000 HZ section.

As shown in Figure 8, Figure 9, and Figure 11, the signals are decomposed by EEMD and then respectively denoised by wavelet hard threshold (WHT), wavelet soft threshold (WST), and WSST. The wavelet semi-soft threshold denoising has the most significant effect on reducing the noise in mid and high frequencies and highest retention of the form of the original signal. Generally, the high SNR and the low RMSE are used as evaluation criteria for the denoising effect. The correlation coefficient is mainly used to distinguish the correlation between each IMF and the original signal, the larger the value, the more correlative components about the original signal that are contained in the corresponding IMF. When the correlation coefficient is above 0.7, the similarity between the denoised signal and the original signal is higher. The evaluation results of the denoising methods are given in Table 3.

The results show that after denoising by the EEMD-WSST, the SNR of the signal is 4.3782, and the RMSE is 1.1196. Compared with wavelet threshold denoising, EMD forced denoising, EEMD forced denoising, EEMD-WHT, and EEMD-WST, the SNR of the signal is higher, and the RMSE is smaller. This shows that the EEMD-WSST signal reconstruction method can achieve a better denoising effect.

## 3. Feature Extraction Based on Multi-Scale Entropy

### 3.1. Calculation Process of Feature Extraction Based on Multi-Scale Entropy

The feature extraction based on multi-scale entropy is to calculate the sample entropy values of the coarse-grained vectors on multiple time scales, which is obtained from the reconstructed signal. The specific calculation steps are as follows:

Step 1. Build coarse-grained vectors. According to the EEMD-WSST signal reconstruction method, the reconstructed signal is $x(t)=\{x_{1},x_{2},\cdot \cdot \cdot x_{N}\}$, and the data length is N. According to the values of pre-given similarity tolerance r and embedding dimension m, the coarse-grained vectors are constructed, as shown in Equation (12).
(12)$$y_{i}^{\tau }=\frac{1}{\tau }\sum_{i=(j-1)\tau }^{j\tau }x_{i},1\leq j\leq \frac{N}{\tau }$$
where τ denotes the scale factor, when τ=1 and $y_{i}^{\tau }=y_{i}^{1}$, that is, the original time series. For the coarse-grained time series of τ>1, the length of each segment is $N^{\prime }=\frac{N}{\tau }$.

The process of coarsening is equivalent to using a sliding window of length τ to calculate the average value of the time series in the sliding window in a non-overlapping way, which is equivalent to using an average filter to remove the high-frequency components of the original time series. The coarse granulation method well reflects the complexity of each scale and can effectively measure the complexity of the time series from multiple scales. The coarse-graining process of the time series at τ=2 and τ=3 are presented in Figure 12.

Step 2. Calculate the distance between $Y^{\left(\tau \right)}(i)$ and $Y^{\left(\tau \right)}(j)$. The absolute value of the maximum Chebyshev distance of the corresponding elements under the scale factor τ is defined as $d^{\left(\tau \right)}[Y^{\left(\tau \right)}(i),Y^{\left(\tau \right)}(j)]$ between $Y^{\left(\tau \right)}(i)$ and $Y^{\left(\tau \right)}(j)$. This can be described by Equation (13).
(13)$$d^{\left(\tau \right)}[Y^{\left(\tau \right)}(i),Y^{\left(\tau \right)}(j)]=\max\left|y^{\left(\tau \right)}(i+k)-y^{\left(\tau \right)}(j+k)\right|$$
where *k* is the integer between 0~m−1, i≠j.

Step 3. Calculate $C_{k}^{\tau ,m}(r)$ and its average value $C_{k}^{\tau ,m}(r)$. For the pre-given similarity tolerance r, count the number of $d^{\left(\tau \right)}[Y^{\left(\tau \right)}(i),Y^{\left(\tau \right)}(j)]<r$ and calculate the ratio of this value to the total number of distances $N^{\prime }-m$, marked as $C_{k}^{\tau ,m}(r)$. Then, calculating the average of $C_{k}^{\tau ,m}(r)$, marked as $C_{k}^{\tau ,m}(r)$. This can be expressed as Equations (14) and (15). Finally, repeat Step 3 for the m-1-dimensional vector to get the $C_{k}^{\tau ,m+1}(r)$ of the m-1-dimensional.
(14)$$C_{k}^{\tau ,m}(r)=\frac{1}{N^{\prime }-m}num\left[d^{\left(\tau \right)}[Y^{\left(\tau \right)}(i),Y^{\left(\tau \right)}(j)]<r\right]$$
where $i,j=1,2,\cdots ,N^{\prime }-m+1;i\neq j$.
(15)$$C^{\tau ,m}(r)=\frac{1}{N^{\prime }-m+1}\sum_{i=1}^{N-m+1}C_{k}^{\tau ,m}(r)$$

Step 4. Calculate the MSE. MSE is defined as the sample entropy at different time scales τ. For each coarse-grained vector, the calculation process of the corresponding sample entropy can be expressed as Equation (16).
(16)$$SE_{\tau }(m,r)=\left|-\ln\frac{C_{}^{\tau ,m}(r)}{C_{}^{\tau ,m+1}(r)}\right|$$

For signal data with length N, its MSE value is presented in Equation (17).
(17)$$MSE=[SE_{1},SE_{2},\cdots ,SE_{\tau }].$$

### 3.2. Influence of MSE Calculation Parameters

The main parameters affecting MSE are embedding dimension m, similarity tolerance r, and scale factor t. The data length N is determined according to the actual length of the collected data, and these parameters need to be determined according to the characteristics and uses of the actual collected data.

(1) The embedding dimension m will affect the amount of information after coarse granulation of the data. The larger the value of m, the greater computational complexity, and the longer the calculation time, the greater requirement of the original data length N. Usually the original data length N needs to be satisfied with $N=10^{m}\sim 30^{m}$. Therefore, generally taking m=2 or m=3. 

(2) The similarity tolerance r will affect the sensitivity of sample entropy to the noise in the reconstructed signal. If r is too small, it will increase the sensitivity of the estimation result to the noise in the reconstructed signal. If r is too large, it will lose a lot of fault information in the time series. The value of r is generally taken as 0.1–0.25 times the standard deviation of the original time series. 

(3) The scale factor τ will affect the length of the coarse-grained vectors. It can be seen in Equation (12) that the original multi-scaling process depends more on the data length, and when τ is larger, the coarse-grained sequences are shorter, which will lead to entropy deviation. The selection of τ needs to be adjusted according to the actual situation. When τ is too large, it will lead to the inability to obtain important information such as fault features in the time series, which will increase the difficulty of calculation. When τ is too small, it does not grasp well the time series as a whole nor express the complexity of the time series. Therefore, it should be selected according to the multi-scale entropy curve, and when the curve gradually tends to be stationary, it should show that the scale factor τ is suitable.

## 4. Fault Diagnosis Model for Rolling Bearings

Based on the advantages of signal reconstruction, the MSE feature extraction technique can retain the signal integrity to the maximum extent on the basis of filtering the high-frequency signal noise and can effectively reflect the complexity of the internal information of the rolling bearing. SVM has the advantages of being able to deal with small samples, non-linear characteristics, and fast classification speed. Therefore, SVM is used to test the classification performance and verify the effect of signal reconstruction and feature extraction methods in this paper. The fault diagnosis model of rolling bearings is as follows:

Step 1. Decompose the signal s(t) by EEMD to obtain the IMFs from high frequency to low frequency. The signal s(t) denotes the vibration signals of different fault types or different degrees of rolling bearings. 

Step 2. Perform the Pearson correlation analysis on each IMF to obtain the Pearson correlation coefficient of each IMF $u=[u_{1},u_{2},\cdots ,u_{n}]$, and then, perform the wavelet semi-soft threshold denoising on the high-frequency IMFs with the Pearson correlation coefficient |u|≥0.7 for denoising.

Step 3. Reconstruct a new eigenvector; $x(t)={\sum_{k=1}^{n}\mathrm{IMF}}_{K}(t)+b(t)$ is reconstructed based on n IMFs after denoising.

Step 4. Perform coarse-graining on the eigenvector $x(t)={\sum_{k=1}^{n}\mathrm{IMF}}_{K}(t)+b(t)$ after signal reconstruction and obtain the coarse-grained vectors, $y_{i}^{\tau }=\frac{1}{\tau }\sum_{i=(j-1)\tau }^{j\tau }x_{i},1\leq j\leq \frac{N}{\tau }$, under the different scale factors, τ. The length of the sample subsequence after coarsening is $M=\frac{N}{\tau }$.

Step 5. Determine the embedding dimension m and similarity tolerance r and calculate the sample entropy of each coarse-grained vector, respectively. Then, obtain the MSE of the eigenvector $MSE=[SE_{1},SE_{2},\cdots ,SE_{\tau }]$.

Step 6. Input the obtained MSE into the SVM model to distinguish the fault types of the rolling bearings.

The flow of the presented model is presented in Figure 13.

## 5. Experimental Verification and Result Analysis

### 5.1. Case 1

The experimental data for rolling bearings analyzed in this study were provided by the Case Western Reserve University Bearing Data Center [29]. The test bearing was a drive-end bearing 6205SKF. The motor speed was 1797 rpm, the faulty diameter of the point faults was 0.021 inches, and the sampling frequency was 12 kHz. In this study, the vibration signals were collected from the inner race fault case, outer race fault case, normal case, and the rolling element fault case, which included four working conditions, one normal case, and three fault cases. Each working condition consists of 50 samples, and each sample had 1024 data points. Among these samples, 40 samples were randomly set as the training data, and the remaining samples were used as test data. The details of the specific data samples are given in Table 4. The obtained time-domain and frequency waveform are presented in Figure 14 and Figure 15.

The rotation frequency corresponding to each type of fault and its fault frequency can be clearly observed from the frequency waveform of Figure 15. According to the bearing fault frequency formula, the rotation frequency $f_{r}$ was 29.95 Hz, the inner race fault frequency $f_{i}$ was 159.72 Hz, the outer race fault frequency $f_{o}$ was 109.83 Hz, and the rolling element fault frequency $f_{b}$ was 78.14 Hz. The basis function selected for wavelet semi-soft threshold denoising was db1. Each sample signal was first decomposed by EEMD. Then the noise in the high frequency of the important IMF, which contains the main fault information, was reduced by wavelet semi-soft threshold denoising. The denoised IMFs were reconstructed to obtain the eigenvector.

Time-domain and frequency waveforms of the reconstructed signals of different fault types after denoising are shown in Figure 16. The correlation analysis method is used to evaluate the correlation between each IMF and the original signal. When the correlation coefficient is greater than 0.7, the similarity between the two signals is higher. After screening, the first two IMFs need to be denoised by the wavelet semi-soft threshold denoising. We can see from Figure 16e that half time rotation frequency, one time rotation frequency, three times rotation frequency, six times rotation frequency, and one time rotation frequency is 31.03 Hz. Figure 16f shows that the inner race fault frequency is 163.16 Hz; Figure 16g shows that the outer race fault frequency is 110.11 Hz, and Figure 16h shows that the rolling element fault frequency is 83.08 Hz. It can be seen that the actual fault frequency is basically consistent with the theoretical calculation frequency. The EEMD-WSST has an obvious denoising effect on the high-frequency part of the signal and can effectively remove the noise in the high-frequency part. The proposed method has more advantages in signal decomposition and can adaptively decompose the signal into an appropriate number of components according to the characteristics of the original signal, and the running speed is faster.

The MSE of the four bearing fault types was respectively calculated, and the scale factor was set as 6. It is unreliable to identify different bearing fault types from the time-domain waveform. Figure 17 presents the MSE curves and scatter diagrams obtained by the EEMD combined with MSE (MSE-EEMD) analysis and the EEMD-WSST combined with MSE (MSE-EEMD-WSST) analysis. Four fault types can be roughly identified from the different MSE values. Figure 17a shows the MSE curves of the reconstructed signal denoised by the EEMD. The rolling element fault case, inner race fault case, and outer race fault case can be clearly distinguished, but the MSE curve of the normal case crosses other MSE curves. Figure 17b shows the MSE curves of the reconstructed signal denoised by the EEMD-WSST. 

The different fault types are easy to distinguish, and the four MSE value curves overlap only at the beginning. Figure 17c shows that, although the fault feature extraction based on the MSE-EEMD is better than the traditional MSE feature extraction, the inner race fault, outer race fault, and normal case cannot be distinguished accurately when the scale factor is low, and the clustering of each fault type is disperse. From Figure 17d, we can see that there is aliasing between the curves of the inner race fault and the outer race fault. However, the clustering of each fault type is relatively concentrated, which means that different fault types can be more easily classified by the EEMD-WSST signal reconstructed combined with MSE feature extraction.

A total test dataset composed of 200 groups of four rolling bearing fault types was collected by the accelerometer, and the collected vibration signal data were intercepted in segments. The sample length of each group was 1024. The statistical results obtained from 10 experiments are presented in Figure 18. To verify the validity of the proposed method, the fault features extracted by six different feature extraction methods are input respectively into an SVM classifier for comparative analysis. Figure 18a shows the classification results of the combination of original fault signals and sample entropy (SE-OFS). The classification accuracy is between 75% and 82.5%, which is lower. 

Figure 18b presents the classification results of the EMD combined with sample entropy (SE-EMD). The classification accuracy is between 80% and 95%. Although the classification accuracy has been improved, the classification results fluctuate greatly. Figure 18c shows the classification results of the EEMD combined with sample entropy (SE-EEMD). The classification accuracy is between 97.5% and 100%. The classification accuracy has tended to be stable, but there still exist classification errors. However, Figure 18d shows the classification result of the combination of original fault signals and MSE (MSE-OFS), and the classification accuracy is between 95% and 100%. 

Compared with SE-OFS, the classification accuracy has been greatly improved. Figure 18e shows the classification result of the EMD combined with MSE (MSE-EMD). The classification accuracy is between 97.5% and 100%, which is almost the same as the SE-EEMD. Figure 18f shows the classification result of the EEMD-WSST combined with MSE (MSE-EEMD-WSST), which has a high classification accuracy of 100%. The overall classification accuracy of using MSE is significantly higher than using SE. Therefore, the fault diagnosis method based on EEMD-WSST signal reconstruction and MSE has good classification results and good generalization.

The average classification accuracy of different methods is given in Table 5. The misclassification of six types of feature sets under 10 statistical results are given in Table 5. Among the 400 samples in SE-OFS, 79 samples are misclassified. In particular, the classification error rate of inner race fault is the highest, and the average classification accuracy is only 80.25%. SE-EMD has a total of 52 misclassified samples, and the number of misclassified samples in both normal and outer race fault is 25. Although the number of misclassified samples is reduced compared with SE-OF, the average classification accuracy is still low, at 86.75%. 

SE-EEMD has a total of five misclassified samples, and the average classification accuracy is 98.75%, which is greatly improved compared with the former two, indicating that the EEMD plays an important role in improving the classification accuracy. MSE-OF has a total of seven misclassified samples, which are distributed in inner race fault and outer race fault. The average classification accuracy is 98.25%, and the number of misclassified samples is nearly 11 times less than the SE-OF. There are only three misclassified samples in MSE-EMD, and only inner race fault is misclassified. The average classification accuracy is 99.25%, which is slightly higher than when using only the SE method. 

MSE combined with EEMD-WSST is the proposed method, as the number of misclassified samples is zero, and the average classification accuracy is 100%. According to the statistical results, it can be found that the average classification accuracy of the SE method is significantly lower than the MSE method, and single-scale SE method is not enough for distinguishing different fault types. Relative to the single-scale method, MSE method has better classification performance. In particular, the proposed method can achieve 100% accuracy. With the proposed method, the noisy signals are eliminated by signal reconstruction, and then the SE values are calculated on multiple scales to obtain better classification results.

Taking into account the results of other techniques tested on the same vibrations signals, the method proposed in this paper is either better than or comparable to other available techniques that usually deploy much more complex algorithms, as can be seen from Table 6.

### 5.2. Case 2

To verify the effect of the proposed method in bearing fault diagnosis under the actual working conditions, the bearing fault simulation experiment is carried out using the rotating machinery test bed. The test bed mainly includes a SGM7J-04AFC6S servo motor with a rated output of 400 W, rated current of 2.5 A, rated torque of 1.27 N, and rated speed of 3000 RPM; YMC122A100 accelerometer with a frequency range of 0.310 kHz; POD-0.6 Kg magnetic powder brake with a rated voltage of 24 V, rated current of 0.81 A, rated torque of 6 N·m, speed at 2000 r/min; plum coupling base and bearing chock, connectors, and fasteners; COCO80 data collector, etc. The test bearing is the 7204C/P5 bearing, and the sampling frequency is 1024 Hz. The specific parameters are shown in Table 7. The experiment uses 200 groups of experimental data, four kinds of bearing states, and 50 groups of each type. The specific structure of the test bed is shown in Figure 19.

The time domain and frequency domain chart of four rolling bearing fault types are presented in Figure 20. Figure 20a shows the vibration signal of the rolling bearing under the normal condition. Figure 20b–d shows, respectively, the vibration signals under the inner race fault, outer race fault, and the rolling element fault. We can see in Figure 20e the one-time rotation frequency, three-times rotation frequency, four-times rotation frequency, five-times rotation frequency, and eight-times rotation frequency. The one-time rotation frequency is consistent with the theoretical calculation of rotation frequency, which is 33.03 HZ. The fault frequency of the inner race fault is 200.20 Hz in Figure 20f, the fault frequency of the outer race fault is 133.13 Hz in Figure 20g, and the fault frequency of the rolling element fault is 75.07 Hz in Figure 20h. The actual bearing fault frequency is almost consistent with the theoretical calculation frequency, but there is still residual noise in the high-frequency part of the signal. 

The time domain and frequency domain chart of the reconstructed signal under different fault cases after denoising by the EEMD-WSST are presented in Figure 21. The normalization method is used to normalize the sample to the range of [0, 1], and the normalized vibration signal is denoised by EEMD-WSST. In Figure 21e, we can see the one-time rotation frequency and the three-times rotation frequency of the normal case. Figure 21f shows the one--time rotation frequency, three-times rotation frequency, and the fault frequency of the inner race fault case. Figure 21g shows the one-time rotation frequency, three-times rotation frequency, five-times rotation frequency, and the fault frequency of the outer race fault case. Figure 21h shows the one-time rotation frequency, three-times rotation frequency, one-third times rolling element failure frequency, and one-time rolling element fault frequency. After the EEMD-WSST signal reconstruction, the signal has lost the high-frequency noise and retains the low-frequency useful information. The MSE values of the four fault cases are calculated, respectively, and the fault is distinguished by the SVM classifier.

From Figure 22, the four faults can be roughly identified from different MSE values. In these four cases, Figure 22a shows that the MSE is used to distinguish the different fault types of bearings after denoising by the EMD. The faults of rolling element, inner race, and outer race can be clearly distinguished in the early stage; however, all kinds of faults overlap in the later stage. Figure 22b shows the MSE curves of the reconstructed signal after denoising by the EEMD-WSST. It can be seen that the outer race fault can be easily distinguished from the other fault cases, and the four MSE curves have less intersection and overlap on different scales, which improves the accuracy of fault classification. Figure 22c shows that the clustering of different fault cases is scattered when EMD is combined with the MSE to extract fault features. Figure 22d shows that the clustering of each fault type is relatively concentrated in different bearing states, and the combination of the EEMD signal reconstruction and the MSE feature extraction is more accurate for different fault cases.

A total of 200 groups of test data of rolling bearing fault state are collected by accelerometer, the collected vibration signal data are intercepted in segments, and the sample length of each group is 1024 points. The fault features extracted by six different feature extraction methods are respectively input to the SVM classifier for comparative analysis, and the results are given in Figure 23. Figure 23a shows the classification results of the SE-OFS. The classification accuracy is between 82.5% and 92.5%, which is lower. Figure 23b shows the classification result of the SE-EMD. The classification accuracy is between 85% and 92.5%. Compared with the former, the number of misclassification errors is reduced. 

Figure 23c shows the classification result of the SE-EEMD. The classification accuracy is between 87.5% and 97.5%, and the classification accuracy is slightly improved compared with the former. Figure 23d shows the classification result of MSE-OFS. The classification accuracy is between 97.5% and 100%. Compared with the classification accuracy of using SE, the classification accuracy is greatly improved. Figure 23e shows the classification result the MSE-EMD. The classification accuracy is between 97.5% and 100%. Although the classification accuracy is the same as the former, the overall classification accuracy has been improved. Figure 23f shows the classification result of the MSE-EEMD-WSST, which has a high classification accuracy of 100%. It can be observed from the statistical chart that the classification accuracy based on MSE feature extraction is significantly higher than that of SE. 

The average classification accuracy of different methods is given in Table 8. The misclassification of six types of feature sets under 10 statistical results can be seen from Table 8. SE-OFS has a total of 45 misclassified samples, and the misclassified samples were in the inner race fault case, with an average classification accuracy of 88.75%. SE-EMD has a total of 42 misclassified samples, and there are more misclassified samples in the normal case and rolling element fault case. Compared with SE-OF, the average classification accuracy is slightly improved, which is 89.5%. 

SE-EEMD has a total of 26 misclassified samples, and the average classification accuracy is 93.5%. Compared with the former two, the number of misclassified samples is significantly reduced. MSE-OFS has a total of three misclassified samples, and the misclassified samples are only in the inner race fault case. The average classification accuracy achieves 98.25%, which is 10% higher than SE-OFS. MSE-EMD has a total of two misclassified samples, which are in the normal case and inner race fault case. The average classification accuracy is 99.05%. Compared with the method using SE, the overall classification accuracy is significantly improved. MSE combined with the EEMD-WSST signal reconstruction is the proposed method, the number of misclassified samples is zero, and the average classification accuracy is 100%.

### 5.3. Discussion

The experimental results indicate that the proposed method is reliable and effective. In comparison to the traditional methods, the EEMD-WSST signal reconstruction has more advantages in distinguishing the different rolling bearing fault types under different conditions. MSE can effectively measure the complexity of signals and had a better performance in our experiments. In comparison to the SE, MSE has better performance on extracting the nonlinear fault features of the vibrational signals. The experimental results show that the classification accuracy is improved upon using MSE values of the vibrational signals after signal reconstruction, compared to directly using the MSE values of the original vibrational signals. Especially for combining with the EEMD-WSST signal reconstruction, the classification accuracy is higher than combining with other signal reconstruction methods. The results of two cases support this analysis. 

## 6. Conclusions

For better feature extraction of the bearing and more accurate identification of the bearing fault types, a method based on the EEMD-WSST signal reconstruction combined with the MSE feature extraction is proposed in this paper. The nonlinear and non-stationary fault signals of bearings are decomposed by the EEMD method, and the IMFs in each state are obtained. The IMFs containing rich fault information are screened by correlation analysis, and the reconstructed signals are obtained after denoising by the WSST denoising. The simulation results indicate that the noise can be effectively filtered by the EEMD-WSST. 

The eigenvectors are obtained by calculating the complexity of the reconstructed signal on multiple time scales by MSE, and the eigenvectors are input into the SVM classifier model to identify fault types. Through the verification of the two groups of experiments, we compared the SE and the MSE and, respectively, combined the original fault signal, EEMD, and the EEMD-WSST. The experimental results, which used CWRU datasets and the measurement dataset from laboratory, show that the signal reconstructed by the EEMD-WSST signal reconstruction can better express the fault information of the rolling bearing. When the basis function and layer numbers of WSST were wrongly chosen, the diagnosis accuracy would to become lower. Meanwhile, compared with the SE, the MSE can better extract and reflect nonlinear fault features. Relative to the MSE of the original vibration signal, the MSE of the reconstructed signal can better suppress the random noise and ensure the integrity of the original signal and accurately measure the complexity of the rolling bearing signal under different time scales. Therefore, the fault diagnosis method based on EEMD-WSST signal reconstruction and the MSE has a better classification performance and has prospects for application in the field of rotating machinery. We intend to apply the method to other datasets collected in the project to further verify the effectiveness and generalization ability of this method. In our future work, we will focus on researching more effective and high-performance fault diagnosis methods for rolling bearings.

## Figures and Tables

**Figure 1 entropy-22-00290-f001:**
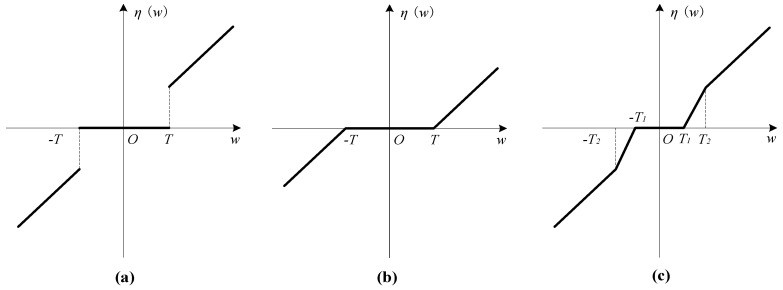
Schematic of three wavelet threshold functions. (**a**) hard thresholding function, (**b**) soft thresholding function, (**c**) semi-soft thresholding function.

**Figure 2 entropy-22-00290-f002:**
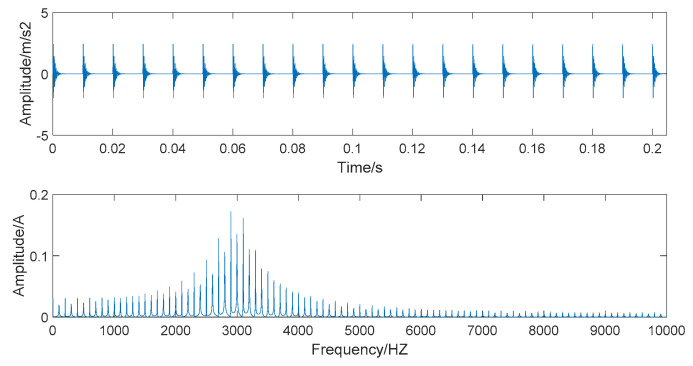
The time domain and frequency domain chart of noiseless simulated signal.

**Figure 3 entropy-22-00290-f003:**
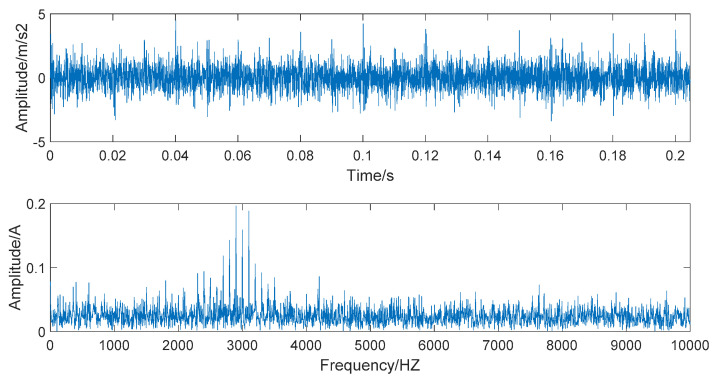
The time domain and frequency domain chart of noise-added simulated signal.

**Figure 4 entropy-22-00290-f004:**
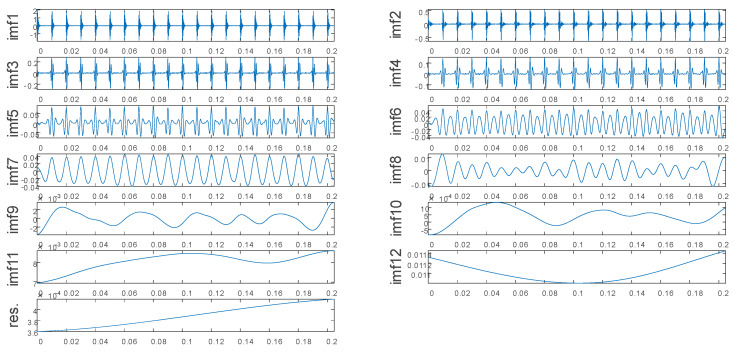
Decomposition result of the noiseless simulated signal.

**Figure 5 entropy-22-00290-f005:**
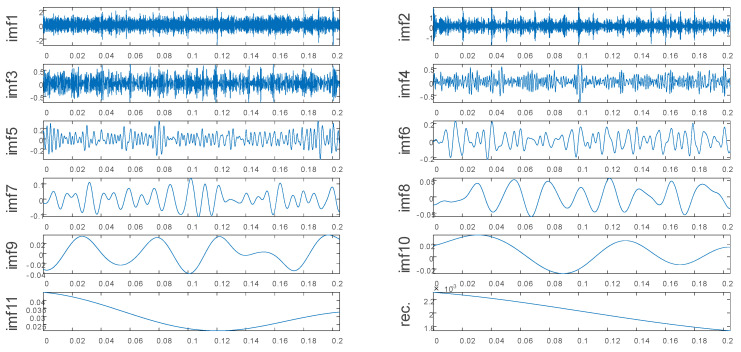
Decomposition result of the noise-added simulated signal.

**Figure 6 entropy-22-00290-f006:**
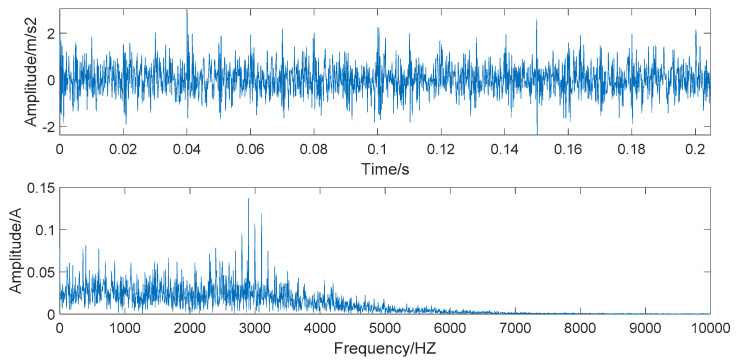
The time domain and frequency domain chart of simulated signal after ensemble empirical mode decomposition (EEMD) forced denoising.

**Figure 7 entropy-22-00290-f007:**
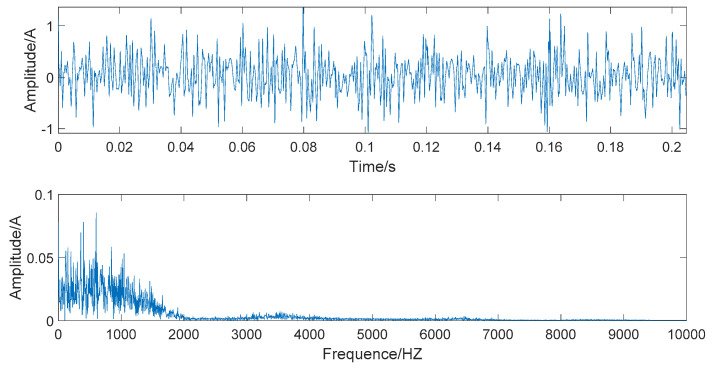
The time domain and frequency domain chart after wavelet threshold denoising.

**Figure 8 entropy-22-00290-f008:**
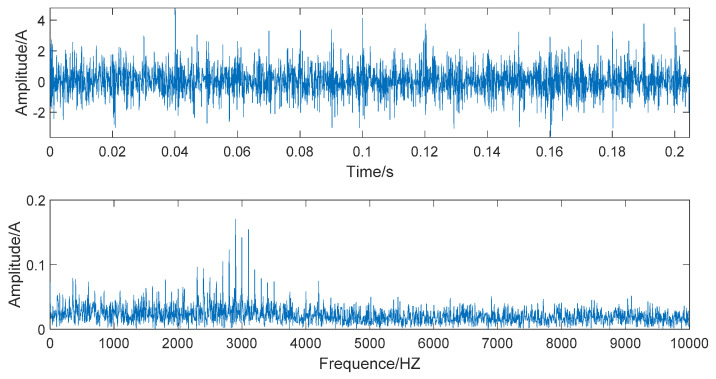
The time domain and frequency domain chart of the signal denoised by the EEMD- wavelet hard threshold (WHT).

**Figure 9 entropy-22-00290-f009:**
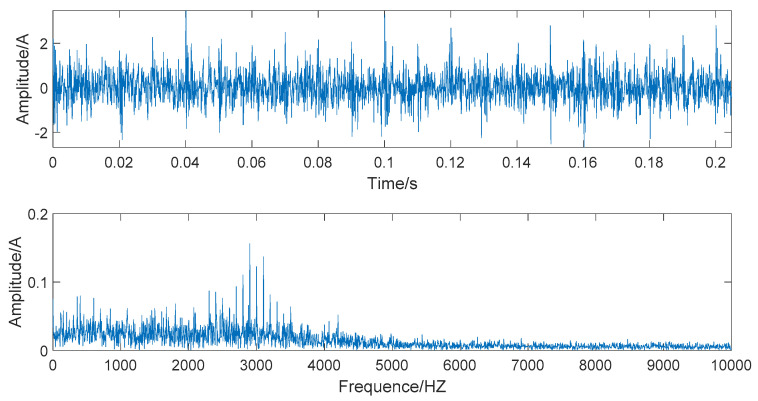
The time domain and frequency domain chart of the signal denoised by the EEMD- wavelet soft threshold (WST).

**Figure 10 entropy-22-00290-f010:**
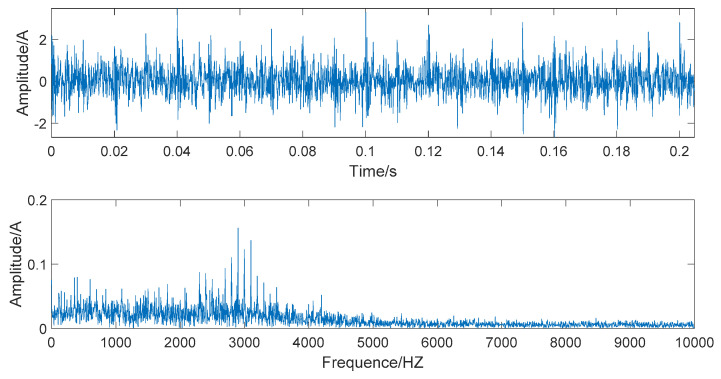
The time domain and frequency domain chart of the signal denoised by the EMD- wavelet semi-soft threshold (WSST).

**Figure 11 entropy-22-00290-f011:**
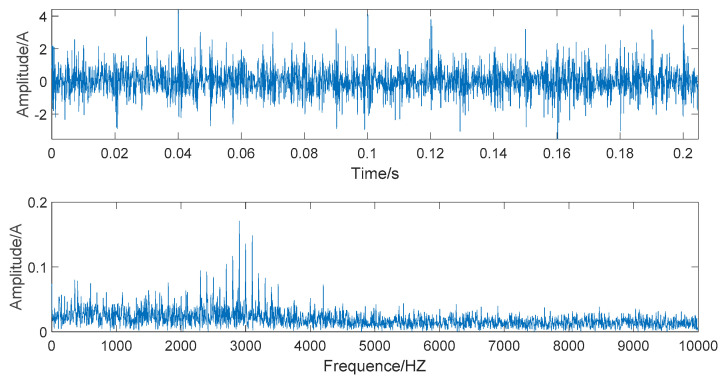
The time domain and frequency domain chart of the signal denoised by the EEMD-WSST.

**Figure 12 entropy-22-00290-f012:**
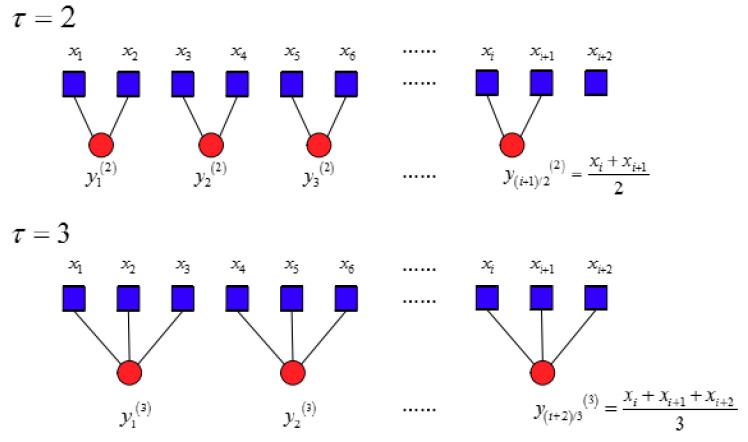
Schematic of the coarse-graining process at τ=2 and τ=3.

**Figure 13 entropy-22-00290-f013:**
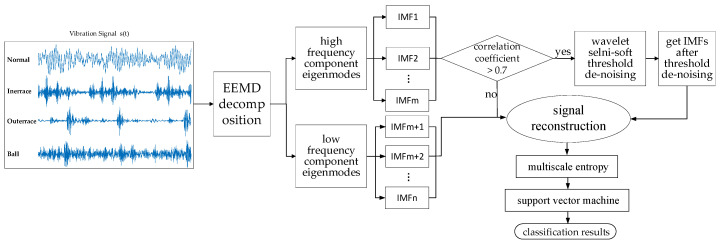
Flow of the presented model.

**Figure 14 entropy-22-00290-f014:**
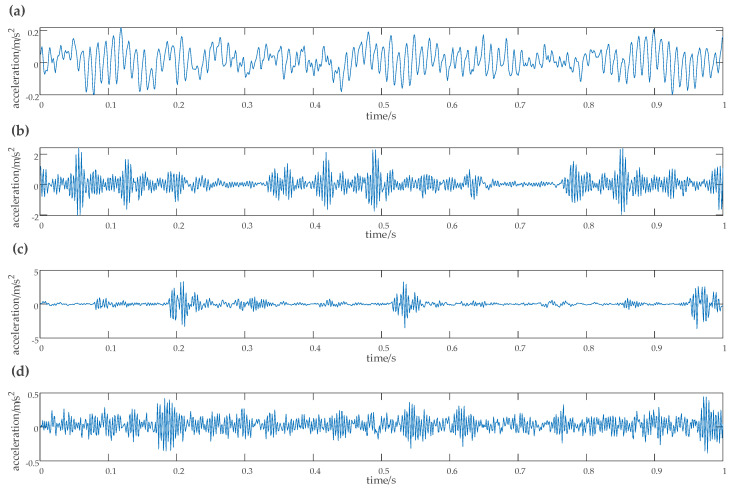
Vibration signal time diagram of the SKF6203 bearing. (**a**) Amplitude of vibration acceleration of the original signal, (**b**) amplitude of vibration acceleration of the inner race fault signal, (**c**) amplitude of vibration acceleration of the outer race fault signal, (**d**) amplitude of vibration acceleration of the rolling element fault signal.

**Figure 15 entropy-22-00290-f015:**
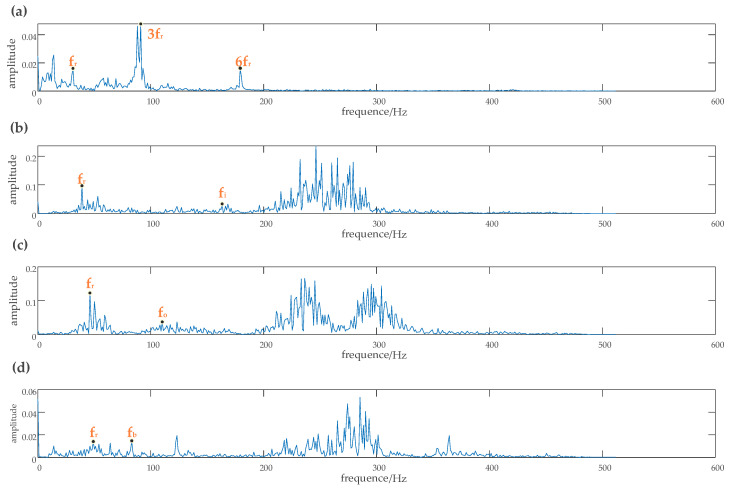
Vibration signal Spectrum diagram of the SKF6203 bearing. (**a**) Spectrum of the raw signal, (**b**) spectrum of the inner race fault signal, (**c**) spectrum of the outer race fault signal, (**d**) spectrum of the ball fault signal.

**Figure 16 entropy-22-00290-f016:**
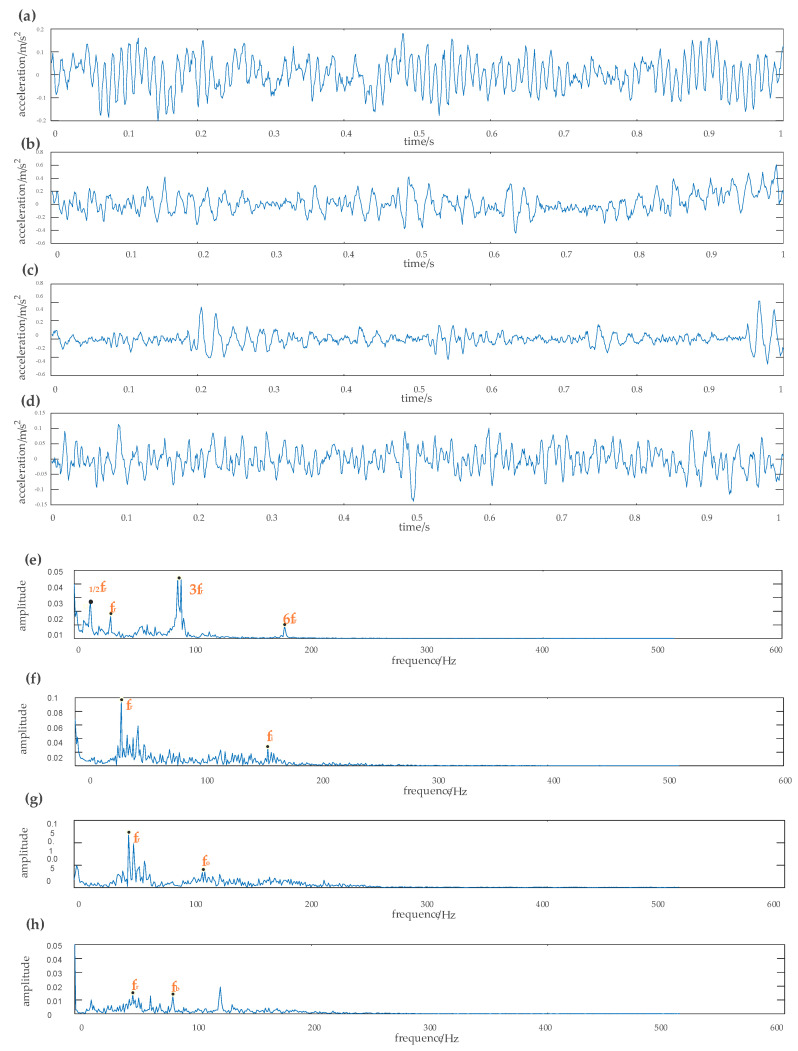
Time-domain and frequency waveform of different signals obtained by EEMD-WSST. (**a**) Amplitude of vibration acceleration of the original signal, (**b**) amplitude of vibration acceleration of the inner race fault signal, (**c**) amplitude of vibration acceleration of the outer race fault signal, (**d**) amplitude of vibration acceleration of the ball fault signal, (**e**) spectrum of the raw signal, (**f**) spectrum of the inner race fault signal, (**g**) spectrum of the outer race fault signal, (**h**) spectrum of the ball fault signal.

**Figure 17 entropy-22-00290-f017:**
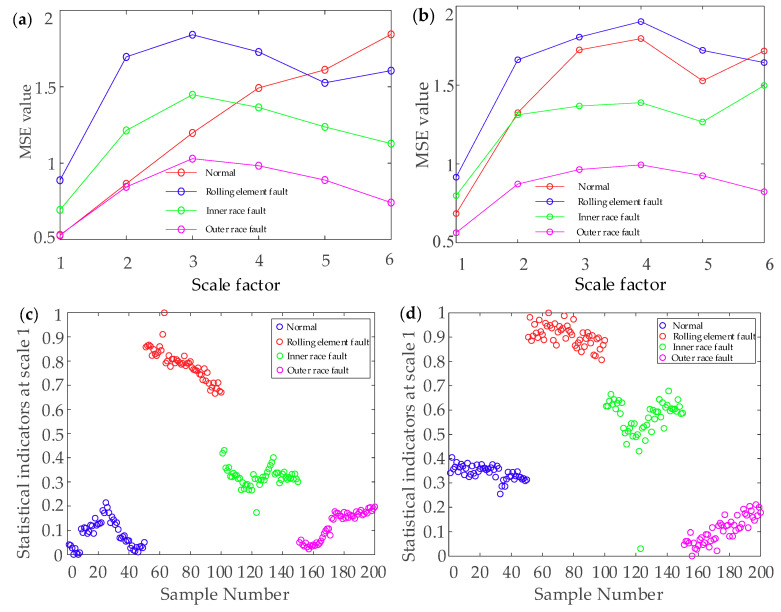
The results of the MSE analysis. (**a**) MSE curve of the MSE-EEMD, (**b**) MSE curve of the MSE-EEMD-WSST, (**c**) MSE scatter diagrams of the MSE-EEMD, (**d**) MSE scatter diagrams of the MSE-EEMD-WSST.

**Figure 18 entropy-22-00290-f018:**
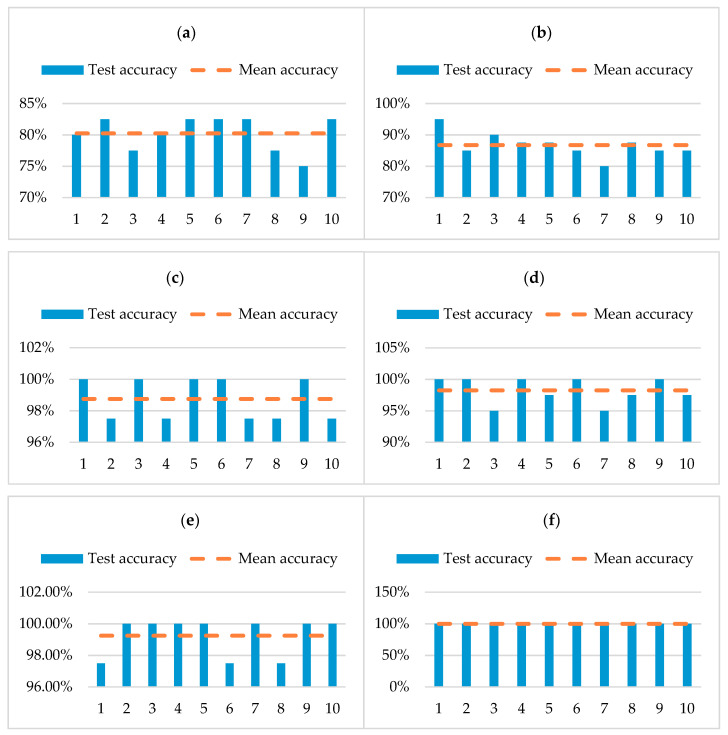
Classification results of fault identification with different characteristics of case 1. (**a**) Classification result of the original fault signals and sample entropy (SE-OFS), (**b**) classification result of the SE-EMD, (**c**) classification result of the SE-EEMD, (**d**) classification result of the MSE-OFS, (**e**) classification result of the MSE-EMD, (**f**) Classification result of the MSE-EEMD-WSST.

**Figure 19 entropy-22-00290-f019:**
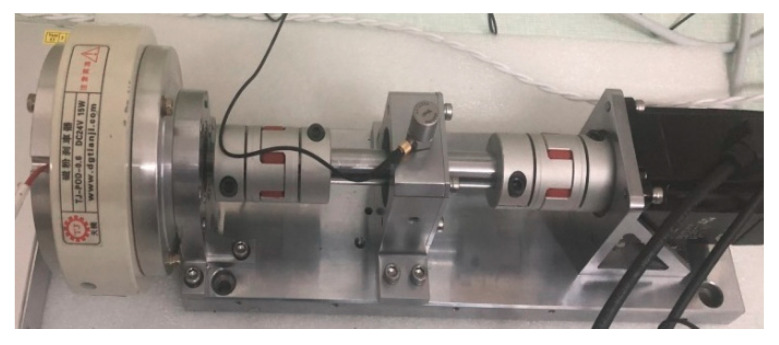
The rotating machinery test bed.

**Figure 20 entropy-22-00290-f020:**
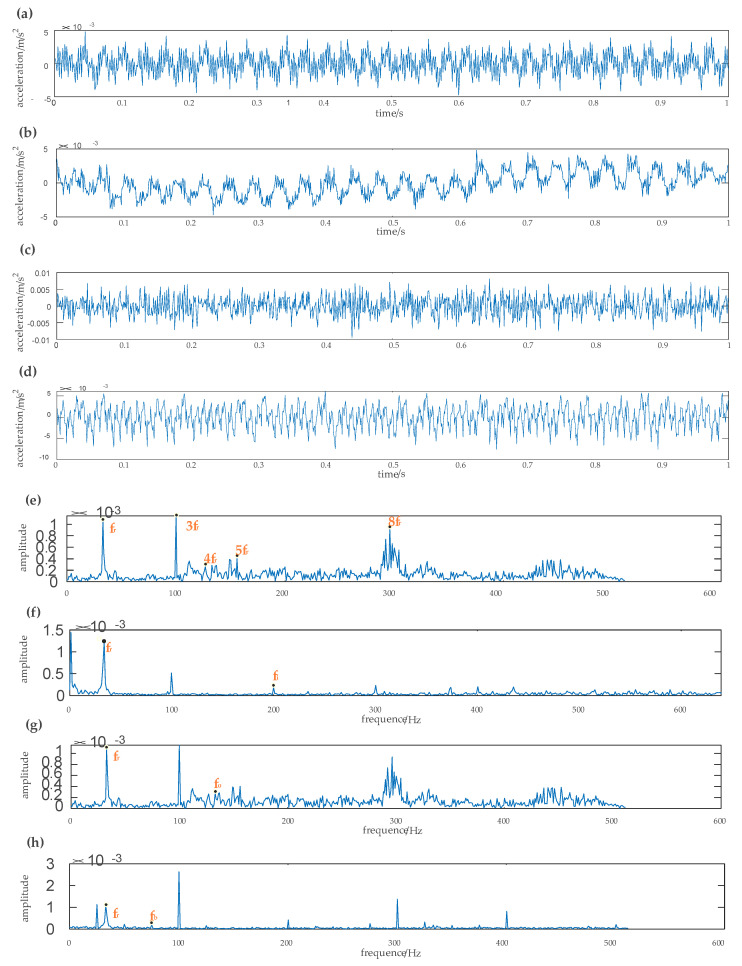
The waveform of the vibration of different fault types. (**a**) Amplitude of vibration acceleration of the original signal, (**b**) amplitude of vibration acceleration of the inner race fault signal, (**c**) amplitude of vibration acceleration of the outer race fault signal, (**d**) amplitude of vibration acceleration of the rolling element fault signal, (**e**) spectrum of the original signal, (**f**) spectrum of the inner race fault signal, (**g**) spectrum of the outer race fault signal, (**h**) spectrum of the rolling element fault signal.

**Figure 21 entropy-22-00290-f021:**
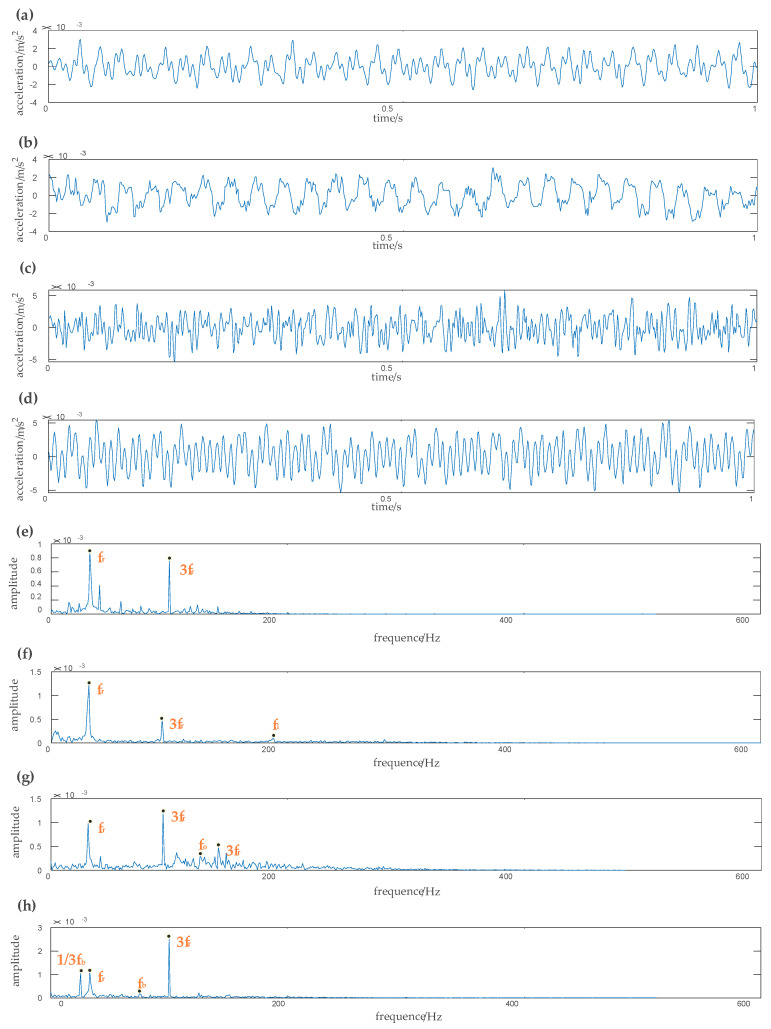
Time-domain and frequency waveform of different signals obtained by EEMD-WSST. (**a**) Amplitude of vibration acceleration of original signal, (**b**) amplitude of vibration acceleration of the inner race fault signal, (**c**) amplitude of vibration acceleration of the outer race fault signal, (**d**) amplitude of vibration acceleration of the rolling element fault signal, (**e**) spectrum of the original signal, (**f**) spectrum of the inner race fault signal, (**g**) spectrum of the outer race fault signal, (**h**) spectrum of the rolling element fault signal.

**Figure 22 entropy-22-00290-f022:**
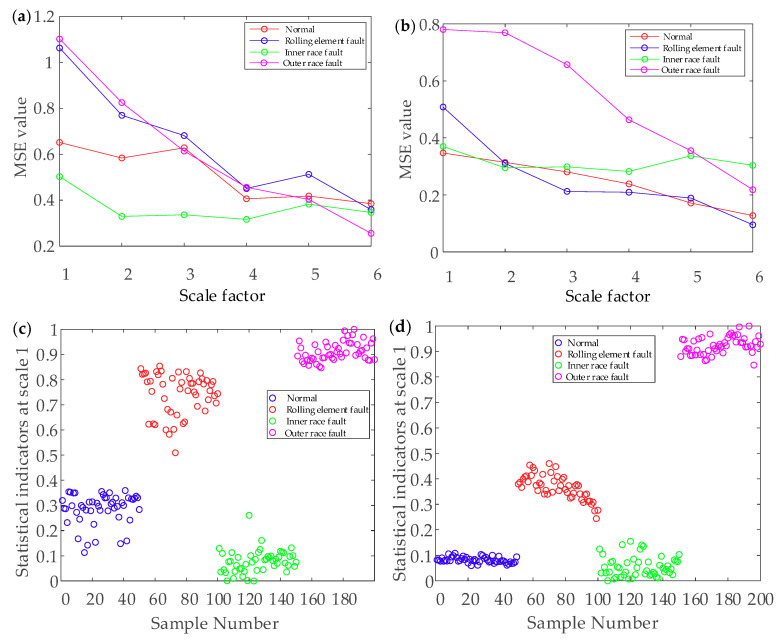
The result of MSE analysis. (**a**) MSE curve of the MSE-EEMD. (**b**) MSE curve of the MSE-EEMD-WSST. (**c**) MSE scatter diagrams of the MSE-EEMD. (**d**) MSE scatter diagrams of the MSE-EEMD-WSST.

**Figure 23 entropy-22-00290-f023:**
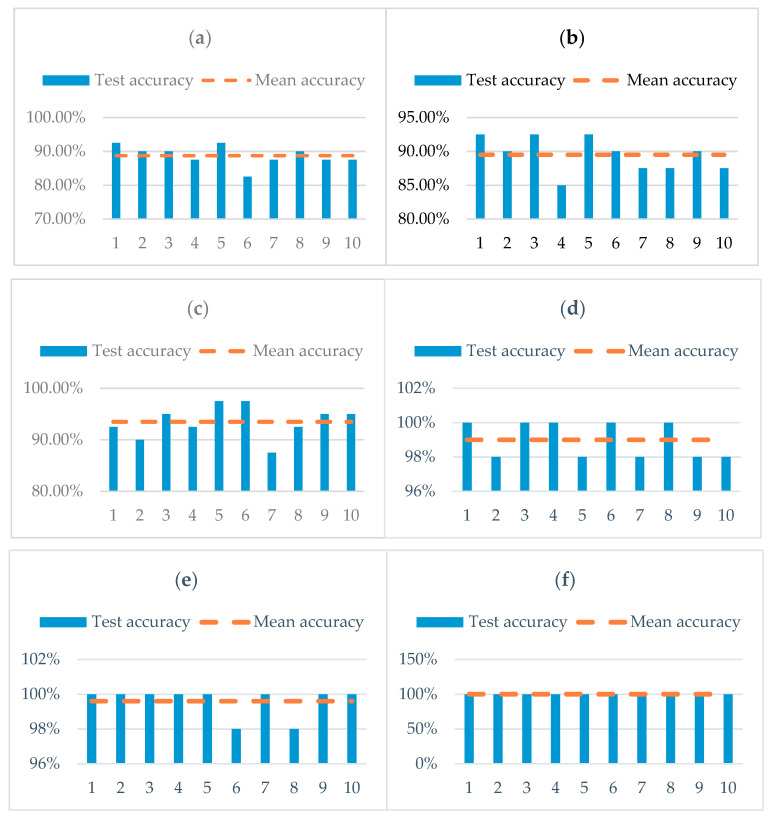
Classification results of fault identification with different characteristics of case 2. (**a**) Classification result of the SE-OFS, (**b**) classification result of the SE-EMD, (**c**) classification result of the SE-EEMD, (**d**) classification result of the MSE-OFS, (**e**) classification result of the MSE-EMD, (**f**) classification result of the MSE-EEMD-WSST.

**Table 1 entropy-22-00290-t001:** Correlation degree based on Pearson correlation coefficients.

Pearson Correlation Coefficient	Correlation Degree
0.7–1.0	strong linear correlation
0.4–0.7	significant correlation
0–0.4	weak linear correlation

**Table 2 entropy-22-00290-t002:** The correlation analysis results of intrinsic mode functions (IMFs).

IMFs	IMF1	IMF2	IMF3	IMF4	IMF5	IMF6	IMF7	IMF8	IMF9	IMF10	IMF11/Rec.	Rec.
**EMD correlation coefficient**	0.7612	0.5694	0.3336	0.2584	0.1726	0.1319	0.092	0.0483	0.0261	0.0310	0.0204	0.0157
**EEMD correlation coefficient**	0.7142	0.531	0.2981	0.1891	0.1469	0.1026	0.0806	0.0387	0.0151	0.0217	0.0112	

**Table 3 entropy-22-00290-t003:** The evaluation results of the denoising methods.

	Wavelet Denoising	EMD Forced Denoising	EEMD Forced Denoising	EEMD-WHT	EEMD-WST	EMD-WSST	EEMD-WSST
RMSE	1.7239	1.3685	1.3560	1.5656	1.5738	1.2203	1.1196
SNR	0.6290	2.6342	2.7145	1.4655	1.4206	3.6300	4.3782

**Table 4 entropy-22-00290-t004:** The details of the four conditions.

Working Condition	Defect Size (Inches)	Number of Training Data Points	Number of Testing Data Points	Label of Classification
Normal	0	40	10	1
Inner race	0.021	40	10	2
Outer race	0.021	40	10	3
Rolling element	0.021	40	10	4

**Table 5 entropy-22-00290-t005:** Fault identification results of different feature extraction methods.

Feature Set	Total	Normal	Inner Race Fault	Outer Race Fault	Rolling Element Fault	Average Classification Accuracy (%)
		Misclassification Number	Misclassification Number	Misclassification Number	Misclassification Number	
SE-OFS	79	4	75	0	0	80.25%
SE-EMD	52	25	1	25	1	86.75%
SE-EEMD	5	0	5	0	0	98.75%
MSE-OFS	7	0	3	4	0	98.25%
MSE-EMD	3	0	3	0	0	99.25%
MSE-EEMD-WSST	0	0	0	0	0	100%

**Table 6 entropy-22-00290-t006:** Comparison between a number of other techniques. Support vector machine (SVM), higher order statistics analysis (HOSA), principal components analysis (PCA), artificial neural networks (ANN), ensemble empirical mode decomposition (EEMD), inter-cluster distance (ICD), multiscale permutation entropy (MPE), stacked sparse denoising autoencoder (SSDAE), fault diagnosis model based on ensemble deep neural network and convolution neural networks (CNNEPDNN), Feature-to-Feature- and Feature-to-Category- Maximum Information Coefficient (FF-FC-MIC), Hilbert-Huang transform (HHT), window marginal spectrum clustering (WMSC).

Reference	Feature Extraction	Classification	Accuracy (%)
[30]	HOSA + PCA	“one-against all” SVM	96.98
[31]	Time–frequency domain	ANN	93.00
[32]	Time- and frequency-domains	SVM	98.70
[33]	IMFs decomposed by EEMD	SVM with parameter optimized by ICD	97.91
[34]	EEMD-MPE	SSDAE	99.60
[35]	CNNEPDNN	CNNEPDNN	98.10
[36]	FF_FC_MIC	SVM	99.17
[37]	HHT-WMSC	SVM	100

**Table 7 entropy-22-00290-t007:** The specific parameters of the 7204C/P5 bearing.

Outer Diameter/mm	Inner Diameter/mm	Pitch Diameter/mm	Ball Number	Ball Diameter/mm	Contact Angle/°	Rotation Frequency/Hz	Inner Race Fault Frequency/Hz	Outer Race Fault Frequency/Hz	Rolling Element Fault Frequency/Hz
20	47	33.5	10	7.4	15	33.33	202.207	131.092	72.01

**Table 8 entropy-22-00290-t008:** Classification results of fault cases under different methods in case 2.

Feature Set	Total	Normal	Inner Race Fault	Outer Race Fault	Rolling Element Fault	Average Classification Accuracy (%)
		Misclassification Number	Misclassification Number	Misclassification Number	Misclassification Number	
SE-OFS	45	0	45	0	0	88.75%
SE-EMD	42	19	2	0	21	89.50%
SE-EEMD	26	0	26	0	0	93.50%
MSE-OFS	3	0	3	0	0	98.75%
MSE-EMD	2	1	1	0	0	99. 05%
MSE-EEMD-WSST	0	0	0	0	0	100%

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
