# Peer review of "A Rolling Bearing Fault Diagnosis Method Based on EEMD-WSST Signal Reconstruction and Multi-Scale Entropy"

_entropy, 2020, doi:10.3390/e22030290_

Round 1

Reviewer 1 Report

In the present study, the authors proposed a new method based on ensemble empirical mode decomposition (EEMD), wavelet semi-soft threshold (WSST) signal reconstruction, and multi-scale entropy (MSE) to improve the accuracy of fault diagnosis. The manuscript is technically sounds. By addressing the following issues, the manuscript will be improved for publication. 1. How this approach by using EEMD is different from EMD? 2. On page 6, the way you identify the correlation degree is so subjective. What about using significance tests for defining? 3. I would encourage the authors to explain more about the novelty of their findings and the limitations of this study. At least to the reader, the current manuscript is not obvious. 4. Why using autocorrelation function to determine the noise component and why selecting IMFs with larger Pearson correlation for WSST? 5. The section 3.1 is needed major revision to avoid many types of errors. 6. Comparing MSE-EMD with MSE-EEMD-WSST is confusing. What do the authors want to uncover? 7. I would suggest the authors to make extra effort to improve the writing of the entire manuscript. For example, the caption of some figure is not clear. There are many typos in the main text. The results in table 3 are not consistent with the explanation. 8. I also suggest the authors to make effort to improve the quality of the figures (layout, resolution and size etc.) and tables.

Author Response

Response to Reviewer 1 Comments

Manuscript ID: entropy-713403Type of manuscript: ArticleTitle: A Rolling Bearing Fault Diagnosis Method Based on EEMD-WSST Signal Reconstruction and Multi-Scale Entropy 

Dear expert:

We are writing the letter to convey my thanks and my major revisions of your comments. We are honored to be reviewed by your comments. Those comments, which make up for our shortcomings of considering less, are very important for enhancing our paper. All authors have read and approved the manuscript. We have carefully taken the comments into account and provided responses to each of the points raised by the you. Some necessary corrections have been made, and all the altered passages have been highlighted in light yellow. We really hope that our improvements can meet your approval. 

Point 1: How this approach by using EEMD is different from EMD?

Response 1:

EMD decomposes the signal into several IMF, with different characteristic scales, which is more regular than the signal before decomposition, thus reducing the non-stationarity of the signal and better extracting the effective information from the signal. However, the traditional EMD denoising method is accomplished by removing the high-frequency IMF components, so it will cause the loss of effective information, and there are modal aliasing and endpoint effects in EMD, which will lead to the subsequent use of WSST to reduce the noise of the signal, which can not effectively remove the noise components of the signal. On the other hand, EEMD makes up for the shortcomings of EMD and suppresses modal aliasing and end effect by adding Gaussian white noise to the original signal on the basis of EMD, which is very critical for the follow-up processing steps and can effectively improve the accuracy of diagnosis.

Point 2: On page 6, the way you identify the correlation degree is so subjective. What about using significance tests for defining?

Response 2:

Pearson correlation coefficient is a more appropriate standard to measure the correlation between two signals. By analyzing the similarity between signals, we can find the effective signal masked by noise, which is also a familiar and commonly used method. In this paper, we analyze the correlation between the IMF component and the original signal, so as to find those IMF components with more noise and reduce the noise. The experimental results show that the performance of the Pearson correlation is well. The significance test may be another effective method. We will try to use the significance test for signal screening in our future work. Thank you for your idea.

Point 3: I would encourage the authors to explain more about the novelty of their findings and the limitations of this study. At least to the reader, the current manuscript is not obvious.

Response 3:

Thank expert for pointing out our deficiencies.

Our research is to improve diagnosis accuracy of the entropy methods for rolling bearing fault diagnose under the complex engineering conditions. The main novel idea in this paper is that we proposed a new signal processing method, which use EEMD and WSST, to improve the quality of the fault features extracted by MSE. With the proposed method, the MSE value is more effective for expressing the fault information when the signal has high noise. We have added some sentences to the Introduce (page 3 line 107).

And the limitation of our research is that the shortcomings of the WSST method existed. The wrong choice of basis function and layer numbers may cause the signal to be over decomposed, some of the valuable information will be removed. We have added some sentences to the Conclusion (page 27 line 723).

Point 4: Why using autocorrelation function to determine the noise component and why selecting IMFs with larger Pearson correlation for WSST?

Response 4:

  1. Because the autocorrelation function can find the effective signal masked by noise by analyzing the similarity between the signal and its multipath signal.
  2. Assuming that the original signal is pure, then several IMF components obtained by EEMD decomposition are pure and highly linearly correlated with the original signal. If there is noise in the original signal, then among several IMF components decomposed by EEMD, the linear correlation between the IMF component with no noise or less noise and the original signal will decrease, and the IMF component with more noise will have a high linear correlation with the original signal, so the IMF component with larger Pearson correlation coefficient needs to be denoised by WSST to eliminate the influence of noise on the following steps.

Point 5: The section 3.1 is needed major revision to avoid many types of errors.

Response 5:

Thank expert for pointing out our deficiencies.

We had improved our manuscript. Some necessary corrections have been made, and all the altered passages have been highlighted in light yellow.

Point 6: Comparing MSE-EMD with MSE-EEMD-WSST is confusing. What do the authors want to uncover?

Response 6:

Thank expert for pointing out our deficiencies.

There is a misspelling. We have changed ‘MSE-EMD’ to ‘MSE-EEMD’.

Point 7: I would suggest the authors to make extra effort to improve the writing of the entire manuscript. For example, the caption of some figure is not clear. There are many typos in the main text. The results in table 3 are not consistent with the explanation.

Response 7:

Thank expert for pointing out our deficiencies.

We have improved the title of Section 2.2, Figure 1, Figure 2, Figure 3, Figure 4, Figure 5, Figure 6, Figure 7, Figure 12, Figure 17, Figure 18, Figure 22, Figure 23, Table1, Table 2 and Table 3. And the explanation of table 3 had also been improved.

Point 8: I also suggest the authors to make effort to improve the quality of the figures (layout, resolution and size etc.) and tables.

Response 8:

Thank expert for pointing out our deficiencies.

We have improved the quality of the figures and tables as well as we can. If necessary, we can provide the original drawing.

The above-mentioned major revisions are responses of your comments. Once again, thank you very much for what you have done. Please accept my sincere thanks. Wish you all the best.

Yours truly,

Professor. Ge, Mr. Yin, Professor. Wang, Dr. Xu, Dr Wei.

Reviewer 2 Report

In the attached filed,The entropy.2020.doc the reviewer copied the conclusion given by the paper. In particular, this reviewer seeks that the authors justify the following statements made in this part of the  conclusion as to the assumed benefits as to the new methodology.

This has been done to a point in this paper, but the reviewer seeks a specific set of accounting of the proofs used to come up with such bold declarations. The reviewer spent time looking the paper over, and while the methodology appears promising, the conclusion should enumerate exactly why such claims are made

It may be a fault in the organization of the paper that the authors felt in the end that the summary was all that was needed. If these claims are true, the authors should state what parts of the paper itself support these claims, so the readers do not have to spend a long time hunting the source of the claims and the presumed proof of these claims

In particular the readers of this document should have the convenience of having the ability to see what parts of the document support each statement made in the conclusions. I.e. cite the particular part of the document for each conclusion. Otherwise it is a needle in a haystack problem

Author Response

Response to Reviewer Comments

Manuscript ID: entropy-713403Type of manuscript: ArticleTitle: A Rolling Bearing Fault Diagnosis Method Based on EEMD-WSST Signal Reconstruction and Multi-Scale Entropy 

Dear expert:

We are writing the letter to convey my thanks and my major revisions of your comments. We are honored to be reviewed by your comments. Those comments, which make up for our shortcomings of considering less, are very important for enhancing our paper.  We really hope that our improvements can meet your approval.

Point :

In the attached filed,The entropy.2020.doc the reviewer copied the conclusion given by the paper. In particular, this reviewer seeks that the authors justify the following statements made in this part of the conclusion as to the assumed benefits as to the new methodology.

This has been done to a point in this paper, but the reviewer seeks a specific set of accounting of the proofs used to come up with such bold declarations. The reviewer spent time looking the paper over, and while the methodology appears promising, the conclusion should enumerate exactly why such claims are made

It may be a fault in the organization of the paper that the authors felt in the end that the summary was all that was needed. If these claims are true, the authors should state what parts of the paper itself support these claims, so the readers do not have to spend a long time hunting the source of the claims and the presumed proof of these claims

In particular the readers of this document should have the convenience of having the ability to see what parts of the document support each statement made in the conclusions. I.e. cite the particular part of the document for each conclusion. Otherwise it is a needle in a haystack problem

Response : 

Thanks expert for pointing out our deficiencies.We have clearly shown in the conclusion the supporting basis for the conclusions of the paper:

The experimental results which used CWRU datasets and the measurement dataset from Key Laboratory of Advanced Manufacturing and Intelligent Technology, Ministry of Education show that the signal reconstructed by the EEMD-WSST signal reconstruction can better express the fault information of the rolling bearing. Meanwhile, compared with the SE, the MSE can better extract and reflect nonlinear fault features. Relative to the MSE of the original vibration signal, the MSE of the reconstructed signal can better suppress the random noise and ensure the integrity of the original signal, and accurately measure the complexity of the rolling bearing signal under different time scales. Therefore, the fault diagnosis method based on EEMD-WSST signal reconstruction and the MSE has a better classification performance and has prospects for application in the field of rotating machinery.   

Thanks very much for what you have done. Please accept my sincere thanks. Wish you all the best.

Yours truly,

Professor. Ge, Mr. Niu, Dr. Xu, Mr. Yin, Professor. Wang.

Reviewer 3 Report

The manuscript presents an algorithm for bearing fault diagnosis based on some well-known techniques. However the combination of all those methods seems to be unique for this particular application. Therefore the authors should be clearer and stress the novelty of their work as well as the real need to have one more method for bearing fault diagnosis in addition to many other already developed and successfully tested. The following comments are given to further improve the manuscript quality:

Avoid lumping references, e.g. 2-5 and similar. Instead summarize the main contribution of each referenced paper in a separate sentence and/or cite the most recent and/or relevant one. The authors should also consider adding there a couple of more recently published results in the field, e.g.

https://www.mdpi.com/1424-8220/16/3/316 https://www.sciencedirect.com/science/article/abs/pii/S0360544216311525

The authors should more clearly explain why they use SVM for their classification problem and why this method turns out to be more accurate than some others (e.g. PCA, classifiers, etc). Have they tested the others? The CWRU bearing test dataset is very popular and freely available on the internet and the accuracy of all methods tested on this dataset is very high. Why did the authors choose this dataset to test their technique and not some other which contain much more data and thus are more realistic? Do the authors have a plan to do it as a part of their future work? If yes, please mention it in the conclusion or somewhere else in the manuscript. It would be nice to include a table which contains the accuracy of other methods and the newly developed one which are all tested on the same dataset, e.g. as it was done here https://www.mdpi.com/1424-8220/16/3/316.

In overall the contribution of the manuscript is not so trivial at all but it needs a minor revision.

Author Response

Response to Reviewer Comments

Manuscript ID: entropy-713403Type of manuscript: ArticleTitle: A Rolling Bearing Fault Diagnosis Method Based on EEMD-WSST Signal Reconstruction and Multi-Scale Entropy 

Dear expert:

We are writing the letter to convey my thanks and my major revisions of your comments. We are honored to be reviewed by your comments. Those comments, which make up for our shortcomings of considering less, are very important for enhancing our paper. We really hope that our improvements can meet your approval. 

Point 1: Avoid lumping references, e.g. 2-5 and similar. Instead summarize the main contribution of each referenced paper in a separate sentence and/or cite the most recent and/or relevant one. The authors should also consider adding there a couple of more recently published results in the field, e.g.https://www.mdpi.com/1424-8220/16/3/316 https://www.sciencedirect.com/science/article/abs/pii/S0360544216311525

Response 1:

Thank expert for pointing out our deficiencies.

The question you mentioned has been modified in the manuscript. According to your suggestion, we have also added two references you mentioned, which are references [6] and [8].

Point 2: The authors should more clearly explain why they use SVM for their classification problem and why this method turns out to be more accurate than some others (e.g. PCA, classifiers, etc). Have they tested the others?

Response 2:

Thank expert for pointing out our deficiencies.

Because SVM is a typical and mature classifier, it is excellent at dealing with small samples and non-linear problems, and the bearing fault diagnosis data is non-linear and the data is a small sample. Therefore, SVM is selected as the classifier. The diagnostic accuracy has been very high in experimental verification. From both theory and practice, it can be concluded that SVM is the most suitable method for fault diagnosis, so no other classifier has been tried.

Point 3: The CWRU bearing test dataset is very popular and freely available on the internet and the accuracy of all methods tested on this dataset is very high. Why did the authors choose this dataset to test their technique and not some other which contain much more data and thus are more realistic? Do the authors have a plan to do it as a part of their future work? If yes, please mention it in the conclusion or somewhere else in the manuscript.

Response 3:

Thank expert for pointing out our deficiencies.

Because the CWRU bearing test dataset is highly recognized in the industry, we choose this dataset as a test of the validity of the fault diagnosis method proposed in the article. As you said, in order to further verify the effectiveness of the method, we set up two experimental verifications, which are the CWRU bearing test dataset and the measurement dataset from Key Laboratory of Advanced Manufacturing and Intelligent Technology, Ministry of Education. The method was verified using multiple experimental datasets, and the results show that both are achieved high diagnostic accuracy. Yes, we have a plan to choose some other dataset that is more realistic to test our technique as a part of our future work. In the conclusion part of the article, we wrote: “We intend to apply the method to other datasets collected in the project to further verify the effectiveness and generalization ability of this method.”

Point 4: It would be nice to include a table which contains the accuracy of other methods and the newly developed one which are all tested on the same dataset, e.g. as it was done here https://www.mdpi.com/1424-8220/16/3/316.In overall the contribution of the manuscript is not so trivial at all but it needs a minor revision.

Response 4:

Thank expert for pointing out our deficiencies.

We have added a table that contains the accuracy of other methods which is all tested on the same dataset,as shown in table 6.

Taking into account the results of other techniques tested on the same vibrations signals,

Table 6. Comparison between a number of other techniques. Support vector machine (SVM), higher order statistics analysis (HOSA), principal components analysis (PCA), artificial neural networks (ANN), ensemble empirical mode decomposition (EEMD), inter-cluster distance (ICD), multiscale permutation entropy (MPE), stacked sparse denoising autoencoder (SSDAE), fault diagnosis model based on ensemble deep neural network and convolution neural networks (CNNEPDNN), Feature-to-Feature and Feature-to-Category- Maximum Information Coefficient (FF-FC-MIC), Hilbert-Huang transform (HHT), window marginal spectrum clustering (WMSC).

Reference

Feature Extraction

Classification

Accuracy (%)

[1]

HOSA + PCA

“one-against all” SVM

96.98

[2]

Time–frequency domain

ANN

93.00

[3]

Time- and frequency-domains

SVM

98.70

[4]

IMFs decomposed by EEMD

SVM with parameter optimized by ICD

97.91

[5]

EEMD-MPE

SSDAE

99.60

[6]

CNNEPDNN CNNEPDNN

98.10

[7]

FF_FC_MIC SVM

99.17

[8]

HHT-WMSC SVM

100

The above-mentioned major revisions are responses of your comments. Once again, thank you very much for what you have done. Please accept my sincere thanks. Wish you all the best.

Yours truly,

Professor. Ge, Mr. Niu, Dr. Xu, Mr. Yin, Professor. Wang.

  1. Saidi, L.; Ali, J.B.; Fnaiech, F. Application of higher order spectral features and support vector machines for bearing faults classification. ISA transactions 2015, 54, 193-206.
  2. Ali, J.B.; Fnaiech, N.; Saidi, L.; Chebel-Morello, B.; Fnaiech, F. Application of empirical mode decomposition and artificial neural network for automatic bearing fault diagnosis based on vibration signals. Applied Acoustics 2015, 89, 16-27.
  3. Zhang, X.; Qiu, D.; Chen, F. Support vector machine with parameter optimization by a novel hybrid method and its application to fault diagnosis. Neurocomputing 2015, 149, 641-651.
  4. Zhang, X.; Liang, Y.; Zhou, J. A novel bearing fault diagnosis model integrated permutation entropy, ensemble empirical mode decomposition and optimized SVM. Measurement 2015, 69, 164-179.
  5. Dai, J.; Tang, J.; Shao, F.; Huang, S.; Wang, Y. Fault Diagnosis of Rolling Bearing Based on Multiscale Intrinsic Mode Function Permutation Entropy and a Stacked Sparse Denoising Autoencoder. Applied Sciences 2019, 9, 2743.
  6. Li, H.; Huang, J.; Ji, S. Bearing fault diagnosis with a feature fusion method based on an ensemble convolutional neural network and deep neural network. Sensors 2019, 19, 2034.
  7. Tang, X.; Wang, J.; Lu, J.; Liu, G.; Chen, J. Improving bearing fault diagnosis using maximum information coefficient based feature selection. Applied Sciences 2018, 8, 2143.
  8. Yu, X.; Ding, E.; Chen, C.; Liu, X.; Li, L. A novel characteristic frequency bands extraction method for automatic bearing fault diagnosis based on Hilbert Huang transform. Sensors 2015, 15, 27869-27893.

Round 2

Reviewer 1 Report

The authors have addressed my comments.